# Manipulation of Host Cell Organelles by Intracellular Pathogens

**DOI:** 10.3390/ijms22126484

**Published:** 2021-06-17

**Authors:** Malte Kellermann, Felix Scharte, Michael Hensel

**Affiliations:** 1Abt. Mikrobiologie, Fachbereich Biologie/Chemie, Barbarastr 11, Universität Osnabrück, 49076 Osnabrück, Germany; mkellermann@uni-osnabrueck.de (M.K.); felix.scharte@uni-osnabrueck.de (F.S.); 2CellNanOs–Center of Cellular Nanoanalytics Osnabrück, Universität Osnabrück, Barbarastr 11, 49076 Osnabrück, Germany

**Keywords:** intracellular bacteria, intracellular parasites, viruses, intracellular lifestyle, vesicular transport, pathogen nutrition

## Abstract

Pathogenic intracellular bacteria, parasites and viruses have evolved sophisticated mechanisms to manipulate mammalian host cells to serve as niches for persistence and proliferation. The intracellular lifestyles of pathogens involve the manipulation of membrane-bound organellar compartments of host cells. In this review, we described how normal structural organization and cellular functions of endosomes, endoplasmic reticulum, Golgi apparatus, mitochondria, or lipid droplets are targeted by microbial virulence mechanisms. We focus on the specific interactions of *Salmonella*, *Legionella pneumophila*, *Rickettsia rickettsii*, *Chlamydia* spp. and *Mycobacterium tuberculosis* representing intracellular bacterial pathogens, and of *Plasmodium* spp. and *Toxoplasma gondii* representing intracellular parasites. The replication strategies of various viruses, i.e., Influenza A virus, Poliovirus, Brome mosaic virus, Epstein-Barr Virus, Hepatitis C virus, severe acute respiratory syndrome virus (SARS), Dengue virus, Zika virus, and others are presented with focus on the specific manipulation of the organelle compartments. We compare the specific features of intracellular lifestyle and replication cycles, and highlight the communalities in mechanisms of manipulation deployed.

## 1. Introduction

Infectious diseases represent a global burden for human health and are still responsible for high levels of morbidity and mortality [1]. Infections are caused by pathogenic bacteria, viruses, or parasites which enter the body through distinct paths. Foodborne pathogens can colonize daily nutrition and consequently enter hosts by consumption of contaminated food or water [2]. Other causative agents are transmitted by mosquitos or ticks that serve as vectors and enter the human blood stream to initiate colonization [3]. Additionally, several pathogens are inhaled in form of infectious droplets originating from, and propagated by infected individuals due to coughing, talking, or sneezing [4].

Following entry, pathogens manipulate host cells to establish an environment that provides nutrition for growth, protection against immune responses, and support spread within the host. Accordingly, various mechanisms of pathogen-induced manipulation have been discovered. A majority of bacterial pathogens deploy effector proteins for host manipulations, marking them as a central element for pathogenesis [5]. Intracellular pathogens rearrange host cell compartments into niches in which efficient proliferation occurs. To do so, however, they have to overcome certain obstacles, starting with cell entry (Figure 1).

There are various possibilities for pathogens to overcome the plasma membrane and invade host cells, both actively and passively. For active invasion, bacterial pathogens translocate effector proteins into the cytoplasm exploiting cytoskeletal elements, or deploy surface-bound invasins to induce entry in form trigger or zipper mechanisms [6]. Passive cell entry results from receptor-mediated endocytosis or phagocytosis. Following invasion, intracellular pathogens localize in endosomes or phagosomes, which they have to manipulate in order to bypass lysosomal degradation [7]. Some pathogens escape the vesicular organelles to replicate inside the host cell cytoplasm, while others take advantage of its vesicular transport along the endocytic pathways to reach target structures. Additionally, there are pathogens that remain inside compartments and redesign these into replicative-permissive environments. In order to achieve replication, viruses need to exploit host cell protein synthesis, thus making the endoplasmic reticulum (ER) a central target for viral pathogens [8]. Further pathogen-induced manipulations of the ER include the redirection of vesicles transported from ER to Golgi containing nutritional cargo. As sorting, modification, and transport of newly synthesized proteins are main Golgi functions, pathogens evolved mechanisms of manipulation to impair and take advantage of these processes to serve proliferation of pathogens [9]. There are pathogens that benefit from manipulating mitochondria to either mitochondrial fusion or fission, showing the great variety of manipulation mechanisms that evolved [10,11]. Accordingly, lipid droplets show to be exploited for lipid supply by bacteria and parasites, while viruses utilize the vesicular organelles for secretion and subsequent cellular spread [12].

## 2. Membrane-Bound Organelles Manipulated by Intracellular Pathogens

### 2.1. Endosomes and Phagosomes

Endocytosis is a process in which extracellular material is internalized by cells through formation of cargo-containing vesicular organelles, the endosomes. Subsequent transport of endosomes to cellular compartments depends on sorting mechanisms mediated by GTPases and can either lead to degradation, modification, or recycling of the content [13]. Phagosomes are intracellular vesicles created by internalization of apoptotic bodies or pathogens. The corresponding process is phagocytosis, which is crucial for cellular immunity and carried out by specialized cells, the phagocytes [14]. Degradation of pathogen-containing phagosomes requires fusion events with lysosomes, another type of cellular organelles containing several hydrolases for molecular degradation. Additionally, lysosomes take part in further degradations including endosomal content or apoptotic cells [15].

Various pathogens rearrange phagosomes into compartments that promote proliferation, or hijack endosomal transport to reach specific subcellular targets. *Salmonella enterica* is a facultative intracellular Gram-negative Enterobacterium causing diseases ranging from self-limiting gastroenteritis to systemic life-threatening typhoid fever [16]. *Salmonella* actively invades non-phagocytic epithelial cells using the *Salmonella* pathogenicity island 1 (SPI1)-encoded type III secretion system (T3SS) that translocates effector proteins mediating trigger invasion [17]. *Salmonella* can also gain access to host cells by phagocytosis. Both entry mechanisms lead to bacteria residing in vesicular compartments that are rearranged during infection into *Salmonella*-containing vacuoles (SCV). The SCV allows bacterial survival and replication, and further remodelling of the host cell endosomal system ensures continuous nutritional supply [18]. 

While phagosome fusion with lysosomes eliminate bacterial agents through enzymatic degradation, many pathogens have evolved manipulation mechanisms to bypass this process. *S. enterica* translocates the SPI1-T3SS effector protein SopB, a phosphoinositide phosphatase that mediates dephosphorylation of phosphatidylinositol-4,5-bisphospate (PI(4,5)P_2_) at the plasma membrane promoting bacterial invasion [7]. Bakowski et al. [19] showed that SopB activity reduced amount of PI(4,5)P_2_ as well as phosphatidylserine (PS) in SCV membranes, resulting in a decrease of the negative membrane surface charge (Figure 2). The authors concluded an impairment of fusion events between phagosomes and lysosomes on the basis of such membrane alterations. Additionally, numerous other pathogens that reside in intracellular vacuoles following invasion such as *Shigella flexneri* show similar membrane alterations to prevent lysosomal fusion [20,21,22]. While S. flexneri escapes from its vacuole to replicate in host cell cytosol and cause severe inflammation, *S. enterica* continues to inhabit the SCV [23]. SCV remodelling to provide nutritional supply requires further expression and translocation of effector proteins of the SPI2-T3SS. Several SPI2-T3SS effectors contribute to formation of an endomembrane network that evolves from the SCV through effector-mediated fusions between endosomes and the SCV. Tubular vesicles interconnect and branch throughout the cytosol, increase fusion events with endosomal vesicles, and thereby increase membrane surface and volume of the vesicular network (Figure 2). *Salmonella*-induced filaments (SIF) form the major portion of tubular structures, and their biogenesis critically depend on function of SifA, with contribution of additional SPI2-T3SS effector proteins [24]. SifA has an N-terminal binding domain for the SifA-Kinesin-Interacting-Protein (SKIP), which interacts with microtubule motor protein kinesin, leading to SIF extension along microtubules throughout the cell (Figure 2) [25,26,27]. Additionally, SPI2-T3SS translocated effector protein PipB2 serves as linker between kinesin and SCV [28]. Functional loss of SifA leads to decreased SCV integrity and subsequent release of bacteria into host cell cytosol [29]. Intracellular *S. enterica* show higher metabolic activity if SIF are formed and connected to SCV compared to mutant strains lacking SIF, underlining the importance of vesicular rearrangements for the pathogen to survive and proliferate in host cells [18].

Viral replication depends on utilization of the host cell biosynthetic machinery for replication of their genome [30]. Following cell entry through receptor-mediated endocytosis, hijacking endosomal transport therefore is pivotal to reach replication sites. There are several viral entry mechanisms including clathrin-mediated endocytosis, macropinocytosis and caveolar endocytosis that lead to pathogens residing in vesicular organelles. As viruses do not alter the endosomal composition to bypass lysosomal degradation, they have to escape or release their genome into the cytosol before fusion events [31]. Enveloped viruses such as Influenza A virus (IAV) mainly relocate their genome into the cytosol through receptor-mediated fusion with endosomal membranes. Influenza viruses are enveloped negative-strand RNA viruses causing mild respiratory diseases, mostly limited to the upper respiratory tract, and in some cases lethal pneumonia [32]. IAV particles contain the protein hemagglutinin that, after activation by organelle acidification, mediates endosomal receptor binding and subsequent membrane fusion [33,34]. In contrast, a majority of non-enveloped viruses such as the positive-strand RNA Poliovirus (PVS)—the causative agent of poliomyelitis—induce conformational changes of the capsid on the basis of receptors interacting with endosomal structures [35]. Following those changes, PVS protein VP4 is liberated into the endosome and functional PVS protein VP1 domains are uncovered to carry out their function in pore formation in the endosomal membrane to promote genome release [36]. The non-enveloped double-strand RNA Reovirus (RV) is released from the endosome as intact particle. Due to membrane-penetrating activities of Reovirus capsid protein µ1 that require protein hydrolysis by endosomal proteases for activation, whole viral particles can escape from the endosome leading to productive infections, inducing apoptosis in cultured cells in vitro, and in vivo heart tissue, and the central nervous system [37,38]. After intracellular replication, many viruses such as the enveloped double-strand DNA Epstein–Barr Virus (EBV) are packed into secretory vesicles leading to exocytosis and cellular spread after assembly [39]. EBV infections mostly appear asymptomatic, but can cause infectious mononucleosis in particular if infection occurs in adolescence or adulthood [40]. Utilization of the secretory pathway by viral pathogens constitutes an additional way for viruses to exploit host cell organelles.

### 2.2. Endoplasmic Reticulum

The endoplasmic reticulum (ER) is among the largest organelles of mammalian cells and consists of a single continuous cytoplasmic membrane that shapes a dynamic network system through its distribution into structural and functional distinct domains [41]. There are two subdomains: The nuclear envelope (NE) and the peripheral ER. Two flat membrane bilayers form the inner and outer nuclear membrane (INM/ONM) which serve as barrier between nuclear and cytoplasmic space [42,43]. Maintenance of the NE requires several interactions such as INM-associated protein binding to chromatin and lamina, linker proteins between INM and ONM for constant spacing, anchoring to cytoskeletal elements, and the positioning of nuclear pore complexes (NPCs) [41,44]. NPCs are essential transport machinery for exchange of selected material through NE and therefore play an important part in regulation of gene expression [45].

There are interconnected tubular and flat sheet membrane structures branching from the NE creating the peripheral ER. Ribosome-free tubular domains of ER membrane (smooth ER) depend on high concentrations of two membrane proteins, RTN and DP1/YOP1. In contrast, there is low abundance of both proteins in membrane sheets that might contribute to the structural curvature of edges [46,47]. Additionally, it was proposed that membrane-bound polyribosomes influence the flat sheet structure forming the rough ER [48]. Several membrane proteins in the ER mediate linkage to microtubules (MT), enabling dynamic changes, e.g., during cell division [41,49]. Further interactions between ER and various other cellular structures indicate additional functions.

Besides protein biosynthesis through translation by membrane-bound ribosomes, regulation of cellular homeostasis in stress situations through induction of the unfolded protein response (UPR), and the modification of translated proteins for further destinations are considered to be main functions of the ER there is also a domain that takes part in lipid biogenesis [50]. The required close proximity to the Golgi apparatus creates the ER-Golgi intermediate compartment (ERGIC), as the organelles interact in distributing mobilized lipids and proteins through the cell [42,48]. Furthermore, the ER serves as store for Ca^2+^ ions and is able to regulate Ca^2+^-dependent signalling pathways such as muscle contraction or apoptosis in various organs [51,52].

The ER is target for manipulations by various intracellular pathogens. *Legionella pneumophila* is a Gram-negative facultative intracellular bacterial pathogen causing Legionnaires’ disease [53]. This pathogen resides in the phagosome after host cell invasion, that is remodelled to an ER-like compartment, termed *Legionella*-containing vacuole (LCV). The LCV enabled bacterial replication, and ultimately fuses with ER membranes for constant nutritional supply. Following invasion, *L. pneumophila* activates expression of virulence genes encoding a type IV secretion system (T4SS) for translocation of effector proteins. By interaction with small GTPases, these effector proteins cooperatively redirect the secretory pathway between ER and Golgi apparatus, leading to fusion of ER-derived vesicles with the LCV (Figure 3 [54,55]. A similar redirection of ER vesicles for bacterial proliferation was reported for the intracellular lifestyle of Gram-negative Proteobacteria *Brucella* spp. and Gram-negative Chlamydiales *Chlamydia trachomatis* [55,56,57].

As host cell regulators for vesicle and membrane transport interact with GTPases RAB1, SAR1 and ARF1, these represent effective targets for translocated effector proteins. Once activated and recruited to the LCV through exchange of GDP to GTP by the effector protein DrrA acting as guanine nucleotide exchange factor (GEF), RAB1 recruits host-tethering factors with important roles in fusion between ER-derived vesicles and LCV (Figure 4). SAR1 is also recruited to the LCV and impacts the budding mechanism of ER-derived vesicles. Experiments interfering with SAR1 function indicated a key role in remodelling the of phagosome [58,59]. ARF1 was proposed as important factor for fusion of the LCV with ER membranes and is activated by the effector protein RalF acting as GEF. After replication in an ER-derived intracellular niche created by manipulation of endosomal pathways, *L. pneumophila* exits its compartment, subsequently the host cell, and spreads to adjacent cells [60]. 

ER is also the organelle most frequently targeted by viral particles after entry [8]. This is due to several circumstances: As one of the largest organelles and by its distribution throughout the cell, the ER can serve after viral rearrangement as structure to efficiently avoid degradation through host cell enzymes such as nucleases. Additionally, the presence of membrane-bound ribosomes for translation of mRNA is convenient for the synthesis of viral proteins. Therefore, viruses with positive-strand RNA such as brome mosaic virus (BMV) shape the peripheral ER to form single-membrane spherule vesicles by presumably modifying ER-associated proteins DP1/Yop1 and reticulons (Figure 3 and Figure 4) [41,61]. Enveloped positive-strand Hepatitis C virus (HCV) and severe acute respiratory syndrome (SARS) virus are pathogens inducing the formation of single-membrane vesicles that can develop into double-membrane vesicles with further progression of infection (Figure 3 and Figure 4). Action of non-structural viral proteins (NSP) drive membrane invaginations. Vesicles originating from ER are partly distinct in structure, but yet provide same function in protection against host cell immune factors, and space for accumulation of components essential for replication [41,62]. Another structural change results from host cell infections by adenoviruses. As this virus influences composition of NPCs through a Kinesin-1-dependent impairment to inject genetic information into the nucleus to reprogram host cells for synthesis of viral elements crucial for their spread [63]. A branched ER membrane network can be observed in cells infected by positive-strand RNA viruses, e.g., *Coronaviridae*. These convoluted membranes are functionally similar to most viral-induced ER rearrangements as they provide space and protection for genome replication and virus assembly [8,64,65].

**Figure 4 ijms-22-06484-f004:**
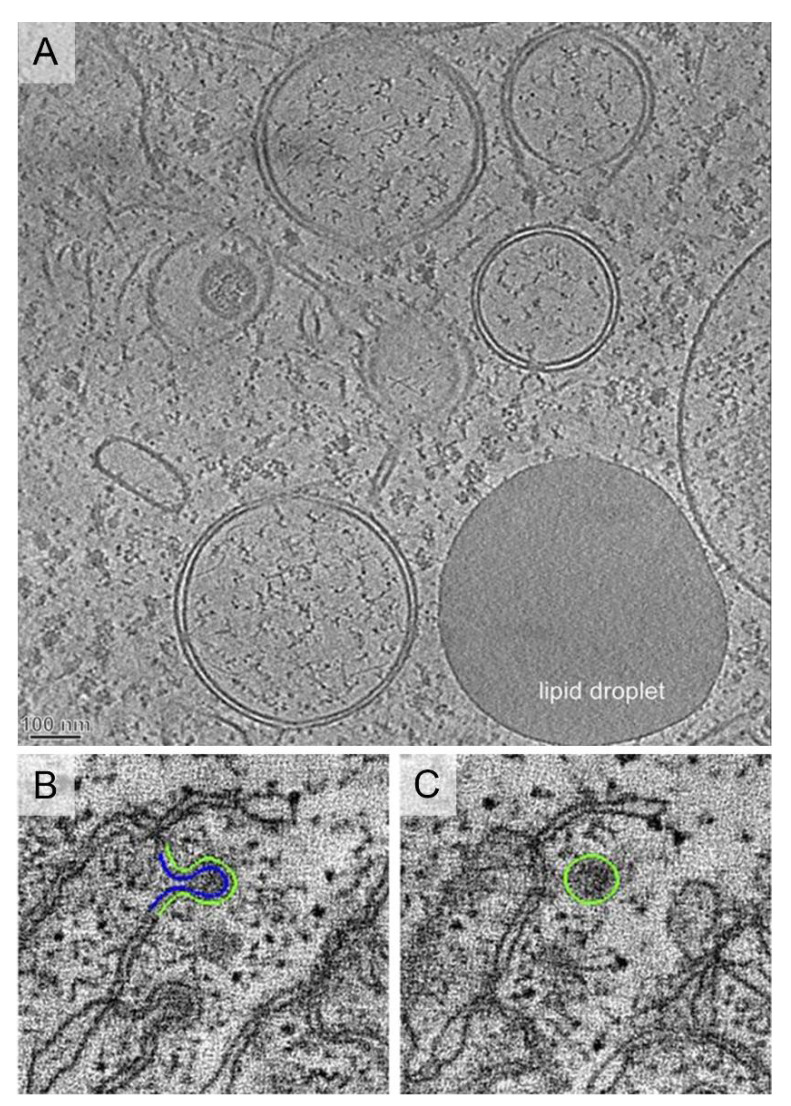
Ultrastructure of viral vesicles. (**A**) Endoplasmic Reticulum (ER)-derived double membrane vesicles (DMV) generated by positive-strand RNA viruses such as Hepatitis C virus (HCV), or severe acute respiratory syndrome coronavirus (SARS-CoV). Image reproduced, with permission, from [62]. Scale bar, 100 nm. (**B**,**C**) Single-membrane spherules formed by brome mosaic virus that localize inside the ER membranes. Images reproduced, with permission, from [66].

The parasite *Toxoplasma gondii* invades nucleated host cells of warm-blooded animals and can cause toxoplasmosis. Similar to a variety of bacterial pathogens, *T. gondii* forms parasitophorous vacuoles (PV), subcellular compartments that function as barrier against host cell immune factors, as well as niche for replication through nutrition obtained from the host cell [67,68]. In contrast to the LCV of *L. pneumophila*, the PV of *T. gondii* does not fuse with vesicles but rather forms narrow contacts with host organelles that might contribute to manipulation of cellular pathways [69]. Experiments showed associations of two organelles with the parasitophorous vacuole membrane (PVM) including the ER. Interactions of proteins secreted by *T. gondii* (GRA3/GRA5) residing in the PVM with the ER-anchoring protein calcium-modulating ligand (CAMLG) might induce association between PVM and organelles, though further studies have to test this hypothesis (Figure 4) [70,71]. It was also shown that the UPR was impaired during *T. gondii* pathogenesis through manipulation of Ca^2+^ efflux from ER, leading to activation of the cytoskeleton-rearranging stress sensor IRE1, promoting host cell migration and parasite dissemination [69].

### 2.3. Golgi Apparatus

The Golgi apparatus is a membrane-bound organelle composed of arrayed flat cisternae that stack and interact with tubular membranes to create a higher-ordered structure named Golgi ribbon [72]. Formation of Golgi ribbons depends on cytoskeletal elements as depolymerization of microtubules results in disruption of the complex structure [73,74]. There are three subdomains within the cell compartment: The cis-Golgi network (CGN) is located in close proximity to the ER and faces its membrane side. The medial-Golgi is the central part of the organelle located between the other two subdomains. The *trans*-Golgi faces the plasma membrane and is able to transform flat cisternae into tubular structures forming the *trans*-Golgi network (TGN) [75,76]. Golgi structures undergo strictly regulated morphological changes during cell cycle that reach as far as disassembly and vesicular packaging of the organelle for mitosis. Reassembly occurs shortly after cell division [77,78].

Post-translational modifications of newly synthesized proteins and of lipids originating from the ER are key roles of the Golgi apparatus. Those modifications are essential for proper transport of cargo to target membranes and functionality, and impaired functions lead to severe diseases [79,80]. The CGN is the receiving end for proteins and lipids synthesized at the ER. Transport vesicles tether and fuse with CGN membranes releasing their cargo into the organelle [81]. There are multiple enzymes throughout the Golgi compartment that perform modifications, such as glycosylation, leading to production of glycoproteins, glycolipids and further modified molecules [82]. Sorting and transport of modified molecules takes place in the TGN and requires diverse sorting signals which are present on cargo and used by coat proteins for correct packaging in vesicles. Additionally, trans-Golgi modifications are done as target membranes need to identify their cognate vesicles [83]. With sorting and transport of molecules participating in host cell immune responses, the Golgi also represents an important organelle for host defence mechanisms against intracellular pathogens. Furthermore, recent studies suggest the Golgi as pivotal element in immune signalling [84].

*Rickettsia rickettsii* is a Gram-negative obligate intracellular bacterium transmitted by ticks causing Rocky Mountain spotted fever, which can be lethal if left untreated [85]. The pathogen encodes a T4SS, as well as several effector proteins which are translocated into the cytosol subsequent to cell entry [86]. Rickettsial ankyrin repeat protein 2 (RARP2) is an effector protein mediating fragmentation of TGN, leading to attenuated vesicular transport and glycosylation defects in host cells (Figure 5) [87]. Studies suggest cysteine protease activity of RARP2 to be essential for TGN fragmentation, though possible target structures remain to be identified [9]. Two proteins are proposed to be affected by glycosylation defects during *R. rickettsii* infections: trans-Golgi protein TGN46 and major histocompatibility complex class 1 (MHC-I). MHC-I is transported to the plasma membrane, functions as antigen presenting complex, and this is crucial for innate and adaptive immune responses against intracellular pathogens [88,89,90]. Transport of MHC-I from TGN to plasma membrane might be impaired in *R. rickettsii* infections, providing protection of the pathogen against host cells innate immune responses, thus enabling pathogen proliferation (Figure 5) [9]. While the CGN is not impaired in infections by *R. rickettsii*, other intracellular bacteria such as *Shigella* disrupt the entire organelle to reduce processes maintaining integrity of the epithelial cell layer [91,92]. 

Disruption of the Golgi apparatus is not exclusively induced by bacterial pathogens, since numerous viruses cause similar effects. Positive-stranded RNA viruses such as Poliovirus (PVS) need to associate with, and rearrange host cell membranes for genome replication [93]. PVS is an Enterovirus of the *Picornaviridae* and causative agent of Poliomyelitis, a severe infectious disease that ranges from mild symptoms to heavy paralysis [94]. Following invasion, the PVS genome is translated into non-structural proteins that differ in function and create an intracellular niche for viral replication. Protein 2B is a viroporin involved in Golgi disruption, as its expression was followed by membrane fragmentation [95]. These membranes are consecutively rearranged into double-membrane vesicles which provide protection against host immune factors, as mentioned above (Figure 5) [96]. The viroporin might support Golgi fragmentation through integration into membranes where it influences Ca^2+^ signalling and promotes permeabilization [97]. Additionally, PVS is able to hinder vesicular transport between the ER and the Golgi, thus inhibiting the secretory pathway. This is due to protein 3A interacting and recruiting GBF1, a GEF for small GTPases ADP ribosylation factor (ARF), to viral vesicles (Figure 5) [98]. Delocalization of GBF1 might lead to lower activation of GTPases that are essential for maintaining the secretory pathway, ultimately leading to reduced translocation of immunity-related molecules such as MHC-I or interleukins [96]. Similar to PVS, HCV induces Golgi disruption to form double-membrane vesicles due to manipulation of GBF1. However, the GEF is not delocalized, but rather phosphorylated mediated by human immunity-related GTPase M (IRGM), that along with GBF1 and ARF, promotes Golgi fragmentation [99]. Taken together, GBF1 appears central for viral replication mechanisms and therefore represents a key target for further investigations [100].

*Plasmodium* spp. parasites cause malaria, leading to 228 million cases globally with 405,000 deaths in 2018 (WHO Malaria report). The parasites are transmitted by mosquitos and possess a complex life cycle with transition between sexual and asexual stages [101]. Humans are infected by sporozoites of the parasite that rapidly enter hepatocytes where they proliferate extensively [102]. Nutritional supply of PV is required so that *Plasmodium* spp. can effectively proliferate, suggesting the existence of host cell manipulation mechanisms. *P. berghei* showed associations of the PVM with Golgi membranes that were maintained throughout the proliferation stage in hepatocytes and followed by Golgi rearrangements to expand membrane interactions which are proposed to improve the parasites nutritional supply. The small GTPase RAB11 is crucial for the organelle’s morphological changes during *P. berghei* infections, as functional mutations diminished this effect [103]. RAB11 takes part in the translocation of trans-Golgi vesicles to cellular target membranes and therefore represents an important element of the secretory pathway [104]. *T. gondii* as well utilizes small GTPases of the Rab family for nutritional supply, indicating common mechanisms of manipulation of secretory pathway elements by intracellular pathogens to promote proliferation [105,106].

### 2.4. Mitochondria

The mitochondrion is a dynamic organelle likely originating from an endosymbiotic α-proteobacterium [107]. Basis for this endosymbiotic theory are similarities in key characteristics of mitochondria and Gram-negative bacteria, e. g. presence of two membranes. The outer membrane separates cell compartments from the cytoplasm and creates an intermembrane space through its distance to the inner membrane. The inner membrane forms structural dynamic cristae due to invaginations and encloses a sub-compartment called the mitochondrial matrix [108,109]. Another indicator for an evolutionary adaptation from an α-proteobacterium is presence of mitochondrial DNA (mtDNA) as circular genome, ribosomes and t-RNAs for autonomous protein translation and protein complexes that might be homologous to those of bacteria [110,111].

Over time, mitochondria acquired several important cellular functions such as synthesis of energy source adenosine triphosphate (ATP) through oxidative phosphorylation, being crucial as disruption or impairment can lead to severe diseases and death in higher eukaryotes [112,113]. As central coordination element for Ca^2+^-dependent pathways inducing intrinsic apoptosis, mitochondria play a key role in regulating programmed cell death. Contact sides to the Ca^2+^-storing ER for material exchange are therefore essential [108,114]. Furthermore, mitochondria release pro-apoptotic proteins such as cytochrome C into the cytosol, following activation of the intrinsic pathway supporting its importance in apoptosis [115,116]. Mitochondria also engage in initiation of immune signalling and take part in innate immune response by producing reactive oxygen species (ROS) that restrict pathogen proliferation and protect the cell from damage [117,118]. Like the ER, mitochondria have a response mechanism to lower mitochondrial stress created through accumulation of unfolded proteins in the matrix that contributes to the regulation of cellular homeostasis in stress situations overall [119]. As a highly dynamic organelle, mitochondria are able to undergo fission and fusion events depending on changing conditions in the cell. Mitochondrial fission is a necessity before cellular division, while fusion events take place when there is a low nutritional supply as it might contribute to maintain energy levels [120,121]. The multifunctional characteristic of the mitochondrion makes it a valuable target for intracellular pathogens. Studies have identified various manipulation mechanisms influencing mitochondria and support understanding pathogenesis of infectious diseases. 

*Chlamydia trachomatis* is an obligate intracellular Gram-negative bacterium responsible for the blinding disease trachoma, as well as for sexually-transmitted diseases of the urogenital tract. It has a reduced genome that lacks important metabolic pathways, explaining the need for hosts to provide nutritional supply [122,123]. *C. trachomatis* differentiates from the infectious form elementary body (EB) to intracellular proliferating form reticulate body (RB). RBs are surrounded by an inclusion membrane (IM) whose associated membrane proteins mediate vesicle recruitment important to ensure continuous nutritional supply [57,124]. The pathogen is able to induce mitochondrial fusion and elongation following invasion. This process requires a phosphorylation of fission promoting dynamin-related protein 1 (DRP1) at serine residue 637 by cyclic AMP-dependent protein kinase (PKA) leading to expansion of the organelle and a higher production of ATP that favours bacterial replication (Figure 6) [10]. Later stages of infections with *C. trachomatis* showed a reversion of elongated mitochondria which might be due to a metabolic shift towards the rapid ATP yielding aerobic glycolysis or Warburg metabolism [125]. 

In contrast, several other pathogens such as *Listeria monocytogenes* or *Shigella flexneri* promote mitochondrial fission following invasion to undermine immune signalling, highlighting the diversity of manipulation strategies that have evolved (Figure 6) [11]. Pores formed by Listeriolysin (LLO) mediate release of *L. monocytogenes* from the PCV. However, LLO also interacts with the plasma membrane, induces Ca^2+^ influx that result in mitochondrial fragmentation [126]. Interactions with apoptotic pathways are an additional characteristic of infections with *C. trachomatis*. The bacterium secretes a protease that degrades pro-apoptotic proteins, thus inhibiting mitochondrial release of cytochrome C and the intrinsic apoptosis pathway. Cellular disintegration is prevented which provides the structural stability necessary for host-dependent pathogens [127,128]. Yet, there are intracellular bacteria like *M. tuberculosis* inducing the opposite effect, leading to host cell disruption for an extensive spread [129]. 

DRP1 is also target for viral pathogenesis revealing manipulation mechanisms affecting mitochondria similar to those of bacteria. In Dengue virus (DENV) infections, mitochondrial elongation through secretion of DRP1 activity-suppressing protein NS4B is used to increase contacts between the organelle and viral membranes. This might support the arrangement of convoluted membranes, ultimately promoting viral replication (Figure 6). DENV is a positive-strand RNA virus of the *Flaviviridae*, causing dengue fever. Additionally, DENV-induced innate immune response was attenuated when elongation of mitochondria occurred [130,131]. Comparable influence on mitochondrial morphology can be observed in viral infections with SARS-Coronaviruses (SARS-CoV) or Zika virus (ZIKV) [130,132]. In contrast to DENV, infections with HCV lead to mitochondrial fission and mitophagy due to a viral protein that activates cyclin-dependent kinase 1 (CDK1). CDK1 subsequently phosphorylates DRP1 at the serine residue 616 promoting its translocation to the mitochondria where it is responsible for fission events and consecutive mitophagy (Figure 6). This might promote viral persistence by restraining mitochondria-mediated apoptosis [133,134]. Many viruses, analogous to *C. trachomatis*, manipulate cellular metabolic pathways so that glucose is used to rapidly produce ATP by aerobic glycolysis instead of the prolonged oxidative phosphorylation presenting a method for immediate supply of energy [135]. 

In addition to ER, mitochondria are further organelles associated with the PVM in *T. gondii*-infected cells [70]. The parasites possess a mitochondrial association factor 1 (MAF1) locus that encodes for several proteins linked to host cell mitochondria association and immune evasion with MAF1b protein as central mediator [136]. Primary reason for association of *T. gondii* with host cell organelles is presumably the need for nutritional supply, allowing the PV to extend [67]. Pernas et al. [137] showed that mitochondrial morphology is indirectly affected during *T. gondii* infection on the basis of a competing counter-mechanism promoting mitochondrial fusion and attenuating uptake of fatty acids by the parasite.

### 2.5. Lipid Droplets

Lipid droplets (LDs) are highly dynamic, lipid-containing organelles that distribute throughout the whole cell performing various functions. LDs form at the ER bilayer where local enzymes synthesize non-charged, neutral lipids such as triglycerides or cholesterol esters whose accumulation between the leaflets of ER membrane leads to lens-like protrusions [138,139]. As consequence of organelle growth through continuous lipid synthesis, LDs bud from the ER membrane into the cytosol. This process requires the activity of fat storage-inducing transmembrane proteins (FIT) which, if functionally disabled, show defects in LD budding [140,141]. Cytosolic LDs structurally distinguish into a hydrophobic core including ER-synthesized neutral lipids, and a phospholipid monolayer that separates cargo from the cytosol [142]. Parallel to cellular distribution, LDs are able to dynamically expand due to lipid-synthesizing enzymes associated with the monolayer. Additionally, various other membrane proteins are located at LDs to mediate transport to, and interactions with organelles such as mitochondria, Golgi, lysosomes, or ER. These interactions are crucial for lipid transfer between both compartments and provide the possibility to counteract starvation-induced energy deprivation of cells as catabolic pathways can break down LD-derived lipids for energy supply [143,144]. Cell growth also requires LD-derived energy supply, as well as further lipids for membrane extensions [145]. LDs represent an important cellular compartment against lipotoxicity since they are dynamic organelles that can not only synthesize and deliver neutral lipids, but also mediate uptake. Likewise, LDs are proposed to temporally store hydrophobic proteins that might accumulate and induce stress-related signalling pathways, thus reducing intracellular stress [142,146]. There are several diseases related to malfunction of lipid storage such as the neutral fat storage disease or obesity, and diverse intracellular pathogens take advantage of LDs to survive and replicate inside host cells [147,148,149]. 

*Mycobacterium tuberculosis* (MTB) is a facultative intracellular pathogen and causative agent of the pulmonary disease tuberculosis (TB) which, if untreated, is often lethal. The bacteria mainly enter host cells through phagocytosis by alveolar macrophages. MTB is able to bypass lysosomal degradation and continuously resides in host cells without inducing strong immune responses, preventing the establishment of clinical symptoms [150,151]. Granuloma formation is the hallmark of TB pathogenesis and can be defined as an accumulation of various immune cells including differentiated and infected macrophages that create the inner core, as well as surrounding T- and B-lymphocytes (Figure 7) [152]. Granulomas can lead to severe tissue damage due to macrophages undergoing MTB-induced necrosis that might promote bacterial replication as necrosis liberates components utilized by MTB [153,154]. Although granulomas are temporally restricting bacterial spread, they also support MTB as protective niche [155]. One specific type of granuloma-associated macrophage acts as key element for TB pathogenesis: the foamy macrophage [156]. Those immune cells are implied to be generated through accumulations of LDs, whose biosynthesis is upregulated as result of peroxisome proliferator-activated receptor γ (PPARγ) expression and activation during MTB infections [157]. Several bacterial genes encode lipid-processing enzymes such as the lipid import system (Mce4) or phospholipase C (PLC), suggesting exploitation of LDs in foamy macrophages by MTB to ensure constant nutritional supply for persistence (Figure 8) [158]. 

LDs are also manipulated by *C. trachomatis*. The bacteria redirect LDs to the IM and subsequently internalize the organelles, likely providing RBs with abundant lipid supply (Figure 8A) [12,159]. Inclusion membrane protein A (IncA), which is required for IM fusion events, is connected with LDs during *C. trachomatis* infections suggesting a pivotal role in LD recruitment [124,160]. There are LD-associated proteins (LDA) expressed by *C. trachomatis*, but their impact on LD manipulation has to be further investigated to understand their functionality [12,151].

The ER-associated transcription factor sterol regulator element-binding protein (SREBP) is essential for signalling pathways that induce lipid synthesis previous to LD biogenesis [161]. Following cell entry, HCV activates SREBP through various molecular mechanisms, resulting in increased lipid and LD synthesis. This benefits HCV as it is able to assemble replicative RNA in lipid-coated particles, or lipoviroparticles (LVPs) (Figure 8B) [162,163]. On the molecular level, HCV encodes NSPs which manipulate lipid homeostasis due to distinct interactions. NSP 5a shows to inhibit the phosphorylation and therefore the regulation activity of adenosine monophosphate-activated protein kinase (AMPK) leading to higher expressions of SREBP [164]. Furthermore, NSP 4b and core protein induce oxidative stress, resulting in the activation of phosphatidylinositol 3-kinase (PI3K), also increasing SREBP expression during HCV infection [163,165]. In DENV infection, association of DENV core protein with LDs is critical since mutations decrease viral replication [166]. These associations might provide a possible way of viral spread, as LDs can be secreted in an autophagy-mediated process. To support autophagy-mediated secretion of LDs, DENV upregulates LD biogenesis through activation of subtilisin kexin isozyme-1/site-1 protease (SKI-1/S1P), which cleaves a precursor of SREBP previous to its nuclear translocation [167,168]. In sum, SREBP is one of the main targets in viral infections that mandatorily require manipulation of lipid synthesis and LDs.

The PV harbouring *T. gondii* associates with host cell organelles for possible pathway manipulations and nutritional acquirements. The parasite might also recruit to PV and subsequently form contacts with LDs. Nolan et al. [169] showed growth reduction of *T. gondii* in mutant host cells depleted of LDs, as well as access to LD content by the parasite. Additionally, the authors confirmed the uptake of LDs into the PV, where lipids can be translocated into LDs of *T. gondii*, resulting in organelle expansion [169]. 

## 3. Conclusions

Mammalian organelles are targets of manipulations by intracellular pathogens. Each intracellular lifestyle features distinct strategies, ranging from simple destruction of an PCV to gain access to host cytosol, to sophisticated redirection of vesicular transport and fusion, resulting in novel compartments that serve as niches for pathogen persistence and proliferation. Recent studies revealed insight into the molecular mechanisms of manipulation, indicated common traits, but also raise questions regarding pathogenesis of diseases caused by many important pathogens. Regardless of similarities and differences, these manipulation mechanisms ultimately serve pathogen survival, replication, and spread. Comparing the mechanisms of manipulation between diverse intracellular pathogens targeting the same organelle could help to understand and therapeutically target recurring virulence patterns in intracellular lifestyles.

## Figures and Tables

**Figure 1 ijms-22-06484-f001:**
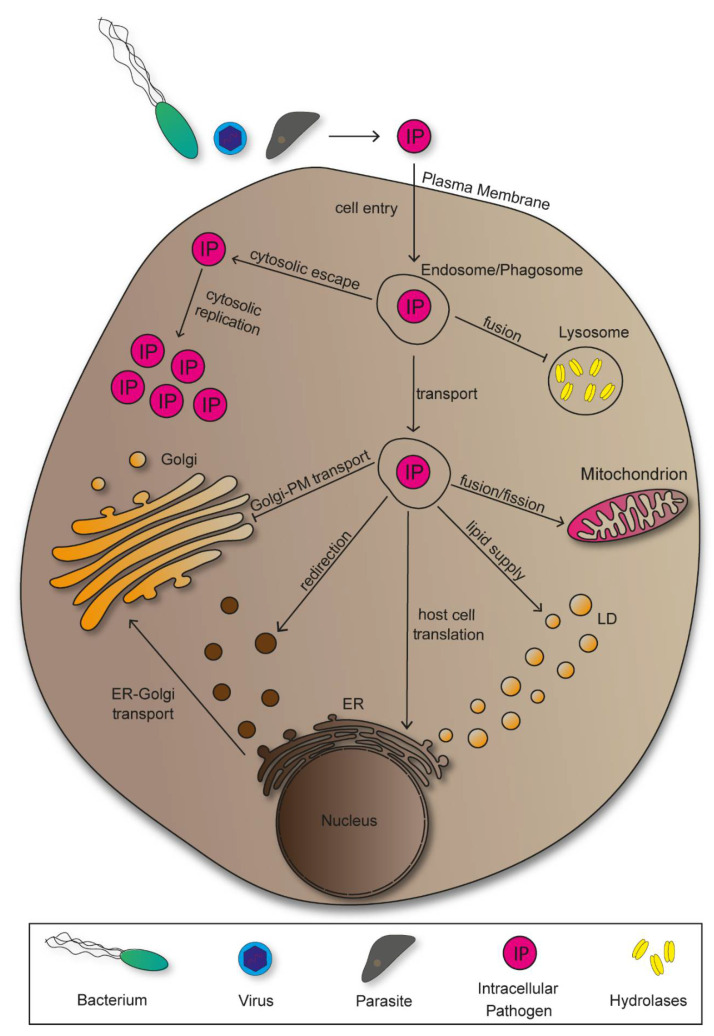
Membrane-bound organelles as targets of manipulation by intracellular pathogens. To initiate infection, intracellular pathogens manipulate the plasma membrane (PM) to enter host cells, or are phagocytosed. Subsequently, intracellular pathogens manipulate endosomes and phagosomes to either escape into cytoplasm, or for transport to further intracellular targets such as the endoplasmic reticulum (ER), the Golgi, mitochondria, or lipid droplets (LDs). The ER is mainly targeted by viral pathogens as they can exploit host cell ribosomes for protein synthesis. Manipulations targeting the Golgi often include disruptions of the secretory pathway. Mitochondria, as energy-generating organelles, are exploited to either decrease or increase cellular energy levels, with both processes promoting proliferation of distinct pathogens. LDs are mainly targeted by pathogens to utilize their lipid content as nutritional supply. However, viruses are able to utilize the vesicles for cellular spread.

**Figure 2 ijms-22-06484-f002:**
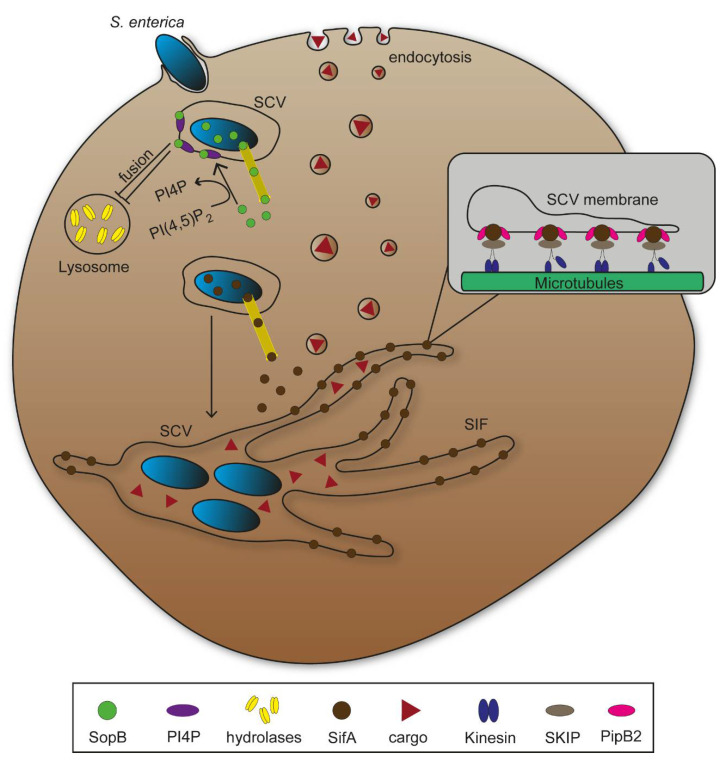
Manipulations of host cell endosomal system by intracellular *Salmonella enterica*. Once internalized, the bacteria rearrange their vesicular compartments to bypass lysosomal degradation. Type III secretion system (T3SS)-secreted effector protein SopB mediates dephosphorylation of membrane lipid phosphatidylinositol-4,5-bisphospate (PI(4,5)P2) generating phosphatidylinositol 4-phosphate (PI4P). Further infection progress shows secretion of effector protein SifA, which associates with the *Salmonella*-containing vacuole (SCV) membrane and leads to *Salmonella*-induced filaments (SIF) formation along microtubules through interactions with SifA-Kinesin-Interacting-Protein (SKIP). SIFs provide nutritional supply through interception of vesicle containing endocytosed cargo such as proteins or lipids resulting in *S. enterica* replication.

**Figure 3 ijms-22-06484-f003:**
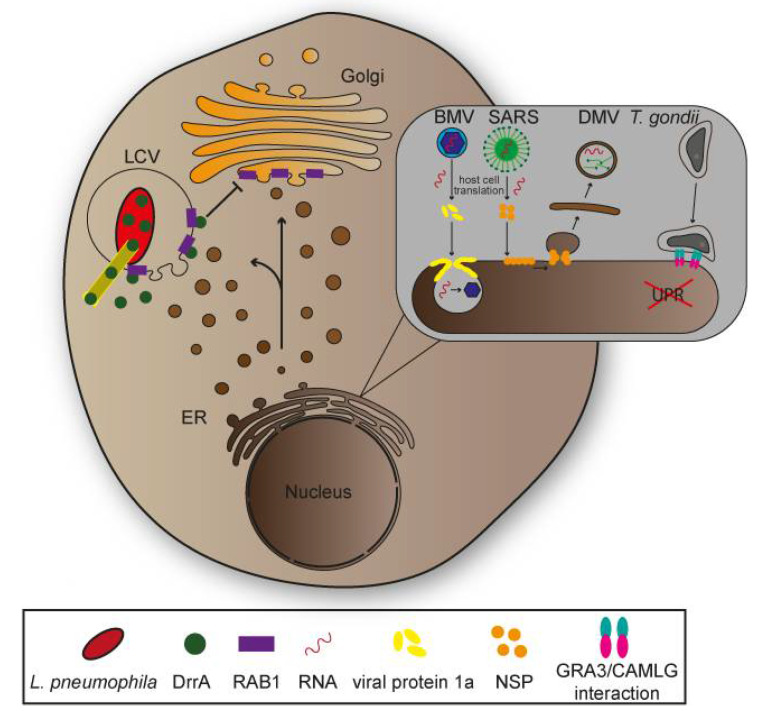
Manipulation of endoplasmic reticulum (ER) by intracellular pathogens. *Legionella pneumophila* recruits small GTPase RAB1 to the *Legionella*-containing vacuole (LCV) membrane through T3SS-secreted effector protein DrrA resulting in redirection of vesicular transport between the ER and the Golgi. *L. pneumophila* thereby ensures nutritional supply for bacterial replication. Brome mosaic virus (BMV) and severe acute respiratory syndrome coronavirus (SARS) exploit the host cell protein machinery to produce viral proteins. BMVs viral protein 1a induces invaginations of ER membranes ultimately creating vesicular spherules suitable for viral replication. SARS non-structural proteins interact with ER membranes and lead to the formation of double membrane ER vesicles (DMV) that promote viral replication. *T. gondii* associates with the ER membrane through interaction protein GRA3 and the ER receptor calcium-modulating ligand (CAMLG) which might support manipulations by *T. gondii* such as inhibition of the unfolded protein response (UPR).

**Figure 5 ijms-22-06484-f005:**
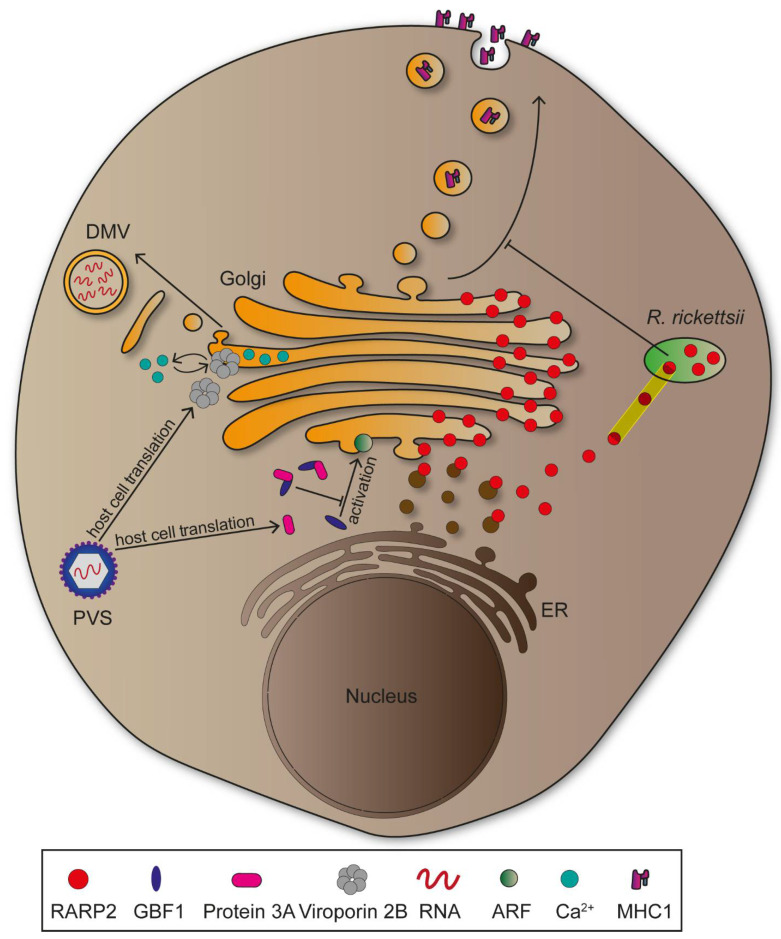
Manipulation of the Golgi by intracellular pathogens. *Rickettsia rickettsii* inhibits transport of major histocompatibility complex 1 (MHC-I) proteins to the plasma membrane through T4SS-translocated effector protein RARP2. RARP2 binds to endoplasmic reticulum (ER)-derived vesicles and relocates to the Golgi after vesicular fusion. It promotes attenuation of trans-Golgi transport by hydrolysing cysteines belonging to a yet unknown protein. Poliovirus (PVS) exploits host cell protein machineries to synthesize Viroporin 2B and Protein 3A. Viroporin 2B forms a pore in Golgi membranes, proposed to result in Ca^2+^ signalling manipulation that promotes double-membrane vesicle (DMV) formation on the basis of Golgi vesicles. Protein 3A relocates the nucleotide exchange factor (NEF) GBF1 which is inhibiting ADP ribosylation factor (ARF) activation on Golgi membranes. Consequently, ER to Golgi transport is inhibited.

**Figure 6 ijms-22-06484-f006:**
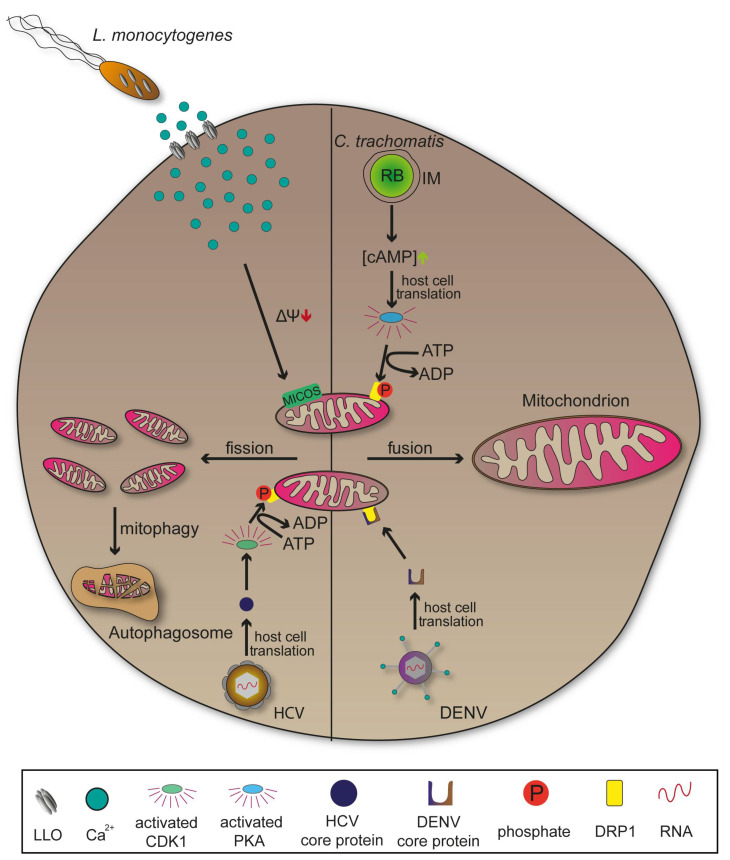
Manipulation of mitochondria by intracellular pathogens. *Listeria monocytogenes* secretes Listeriolysin O (LLO) that forms pores in the host cell plasma membrane leading to Ca^2+^ influx. Subsequent decrease in mitochondrial membrane potential and mitochondrial contact site and cristae organizing system (MICOS) functionality are elemental for mitochondrial fission. *C. trachomatis* exploits the host cell translation machinery to synthesize cyclic adenosine monophosphate (cAMP) enriching proteins whose consequences activate cyclic AMP-dependent protein kinase (PKA). PKA specifically phosphorylates dynamin-related protein 1 (DRP1) to inhibit its function resulting in elongated mitochondria. Dengue virus (DENV) as well inhibits DRP1 function. Hepatitis C virus (HCV) activates the DRP1 phosphorylating cyclin-dependent kinase 1 (CDK1) through core proteins ultimately promoting mitochondrial fission and mitophagy.

**Figure 7 ijms-22-06484-f007:**
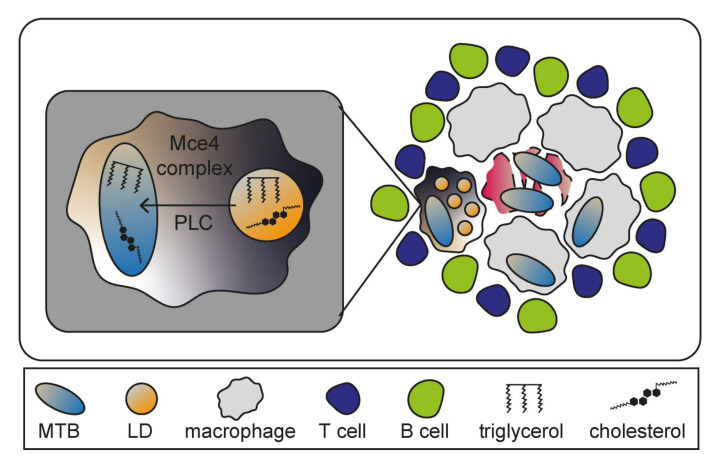
Granuloma formation induced by *Mycobacterium tuberculosis* (MTB). After phagocytosis, MTB mediates the recruitment of diverse macrophages that serve as protection against lymphocytes such as B cells and T cells. This accumulations of immune cells as consequence of MTB infection are termed granuloma. Foamy macrophages have a specialized morphology due to high amounts of lipid droplets (LDs). MTB exploits these LDs for nutritional supply with the help of lipid import systems such as Mce4 complex and phospholipase C (PLC). Severe tissue damage can be observed as result of MTB-induced necrosis of macrophages.

**Figure 8 ijms-22-06484-f008:**
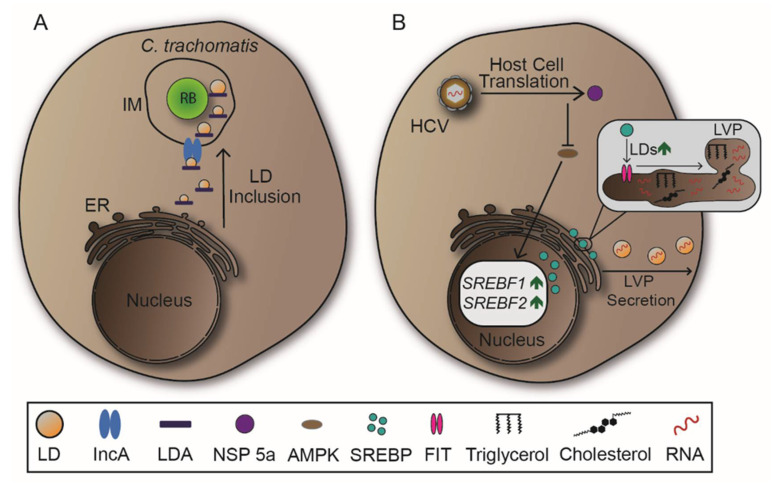
Manipulation of lipid droplets (LDs) by *Chlamydia trachomatis* and Hepatitis C virus (HCV). (**A**) *C. trachomatis* redirects LDs originating from ER to the inclusion membrane through expression of lipid droplets-associated proteins (LDA) and inclusion membrane protein A (IncA). IncA mediates homotypic fusions at the inclusion membrane (IM) and is suggested to mediate the same effect on LDs. Specific LDA functions are yet to be unravel. (**B**) HCV exploits host cell translation machineries to synthesize non-structural protein (NSP) 5a which inhibits phosphorylation and therefore activation of regulatory enzyme adenosine monophosphate-activated protein kinase (AMPK). Subsequently, expression of genes encoding lipid metabolism regulator sterol regulator element-binding protein (SREBP) are increased, leading to higher biogenesis of LDs. HCV uses LDs to distribute its RNA throughout the organism as lipoviroparticles (LVPs) that can be secreted by cells to promote viral spread.

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
