# Peer review of "Manipulation of Host Cell Organelles by Intracellular Pathogens"

_ijms, 2021, doi:10.3390/ijms22126484_

Round 1

Reviewer 1 Report

Kellermann et al. present a comprehensive overview about interactions of intracellular pathogens with different organelles of the host cell. This organelle-centered view illustrates nicely the different strategies of intracellular pathogens to survive and replicate in host cells. Although the manuscript discusses bacteria, viruses and eukaryotic parasites, most if the manuscript concentrates on bacteria, which are discussed extensively, especially Salmonella. For the general readership of the International Journal of Molecular Sciences, I recommend a more in-depth description of viruses and parasites, as well. For example, information on the kind of virus (RNA or DNA virus) and what kind of disease the virus is causing would be helpful. Furthermore, the manuscript is – I am sorry that I have to say this – quite sloppy! There are many inconsistencies and the reference list is not properly formatted in the text.

Additional specific points:

The figures are not mentioned in the text. Please cite where appropriate.

Labeling in the figures is not consistent. Labels are sometimes capitalized, sometimes not. Please harmonize. This applies also to the figure legends (e.g. Figure 3: “A) endoplasmic reticulum” vs. “B) Single-membrane spherules”)

In addition, the text in the figures is slightly fuzzy. This should be improved if possible.

The manuscript would benefit from language editing. At least, the authors should carefully go through the entire text and correct mistakes. Here just some examples from page 1:

Lane 6: “Pathogenic Intracellular bacteria” should read “Pathogenic intracellular bacteria”

Lane 26: “…can colonize daily nutrition and are consequently enter hosts…” should read “…can colonize daily nutrition and consequently enter hosts…”

Lane 39/40: “ultimately promote the pathogens survival” should read “ultimately promote the pathogen’s survival”

Lane 58: Salmonella enterica should be in italics.

Lanes 216 to 218: “After replication…” The sentence is unclear. Please rephrase.

Lanes 231 to 234: “Vesicles originate…” Here, it is unclear what the authors want to say, as well. Please rephrase.

Lane 305: Rickettsia ricketsia should read Rickettsia rickettsia.

Lanes 322 ff.: The part on EPEC should be removed. EPEC are not intracellular pathogens and, thus, do not fit to the topic of this review (see title!).

Lane 363: IRGM is a GTPase and not a kinase. Are the authors sure that IRGM directly phosphorylates GBF1?

Lanes 418+419: “In this lifecycle…” The sentence is unclear. Please rephrase.

Lane 431: “(LLo) that forms porin…” should read “(LLo) that forms pores…”

Author Response

Thank you for the constructive criticism to the initial version of our manuscript. We have taken the points raised into consideration and revised the manuscript accordingly. Please find our response the specific points given below.

Comments and Suggestions for Authors

Kellermann et al. present a comprehensive overview about interactions of intracellular pathogens with different organelles of the host cell. This organelle-centered view illustrates nicely the different strategies of intracellular pathogens to survive and replicate in host cells. Although the manuscript discusses bacteria, viruses and eukaryotic parasites, most if the manuscript concentrates on bacteria, which are discussed extensively, especially Salmonella. For the general readership of the International Journal of Molecular Sciences, I recommend a more in-depth description of viruses and parasites, as well. For example, information on the kind of virus (RNA or DNA virus) and what kind of disease the virus is causing would be helpful. Furthermore, the manuscript is – I am sorry that I have to say this – quite sloppy! There are many inconsistencies and the reference list is not properly formatted in the text.

Response: It appears that the version of the manuscript provided to reviewers had some formatting issues. We apologize if this caused inconvenience, but this is probably an error of the editorial system. Please note that all Figures have been referred to in the text, and that citations were correctly formatted in the version we provided.

We added further information the on the genome structure of the viruses described and give a brief introduction of the related diseases.  

Additional specific points:

 The figures are not mentioned in the text. Please cite where appropriate.

Response: All figures are cited at the appropriate position in the text

Labeling in the figures is not consistent. Labels are sometimes capitalized, sometimes not. Please harmonize. This applies also to the figure legends (e.g. Figure 3: “A) endoplasmic reticulum” vs. “B) Single-membrane spherules”)

Response: We addressed this point accordingly

In addition, the text in the figures is slightly fuzzy. This should be improved if possible.

Response: Please refer to the high-resolution figures provided.

The manuscript would benefit from language editing. At least, the authors should carefully go through the entire text and correct mistakes. Here just some examples from page 1:

Lane 6: “Pathogenic Intracellular bacteria” should read “Pathogenic intracellular bacteria”

Lane 26: “…can colonize daily nutrition and are consequently enter hosts…” should read “…can colonize daily nutrition and consequently enter hosts…”

Lane 39/40: “ultimately promote the pathogens survival” should read “ultimately promote the pathogen’s survival”

 Lane 58: Salmonella enterica should be in italics.

 Lanes 216 to 218: “After replication…” The sentence is unclear. Please rephrase.

 Lane 305: Rickettsia ricketsia should read Rickettsia rickettsia.

Response: The text has been carefully edited, and the instances mentioned above and several others were corrected.

 Lanes 322 ff.: The part on EPEC should be removed. EPEC are not intracellular pathogens and, thus, do not fit to the topic of this review (see title!).

Response: o.k., removed, Figure adjusted

 Lane 363: IRGM is a GTPase and not a kinase. Are the authors sure that IRGM directly phosphorylates GBF1?

Response: According to Ref. 96 the phosphorylation is mediated by IRGM, thus further factors could by involved. We changed to ‘….phosphorylated mediated by…’

 Lanes 418+419: “In this lifecycle…” The sentence is unclear. Please rephrase.

Response: Sentence has been rephrased

 Lane 431: “(LLo) that forms porin…” should read “(LLo) that forms pores…”

Response: o.k., corrected

 Lanes 231 to 234: “Vesicles originate…” Here, it is unclear what the authors want to say, as well. Please rephrase.

Response: Sentence has been rephrased

Reviewer 2 Report

The writing is very clear and this review covers an interesting topic.

My major concern is the scope. The topic covered could fill a couple of books. This has necessitated that the authors cover topics with an incredible amount of variability, which really begs the question as to whom the review is directed at. The basic structure is fine and I enjoyed the little introductions to each organelle. I think the review would be greatly improved by narrowing the scope, perhaps closer to the authors' interest. The majority of microbiologists who read this will wonder why their area interest is missing, and non-microbiologists will find that no topic is covered. There are a few random examples relevant to each organelle. For example, pretty much every virus interferes with mitochondria through a range of mechanisms.

Either the scope should be narrowed (just bacteria and the ER, Golgi, for example) or more needs to done to justify why the specific examples were selected.

In the version I received, there were a significant number of references that were not formatted:

 (Error! Reference source not found.)

Author Response

Thank you for the constructive criticism to the initial version of our manuscript. We have taken the points raised into consideration and revised the manuscript accordingly. Please find our response the specific points given below.

Comments and Suggestions for Authors

The writing is very clear and this review covers an interesting topic.

My major concern is the scope. The topic covered could fill a couple of books. This has necessitated that the authors cover topics with an incredible amount of variability, which really begs the question as to whom the review is directed at. The basic structure is fine and I enjoyed the little introductions to each organelle. I think the review would be greatly improved by narrowing the scope, perhaps closer to the authors' interest. The majority of microbiologists who read this will wonder why their area interest is missing, and non-microbiologists will find that no topic is covered. There are a few random examples relevant to each organelle. For example, pretty much every virus interferes with mitochondria through a range of mechanisms.

Either the scope should be narrowed (just bacteria and the ER, Golgi, for example) or more needs to done to justify why the specific examples were selected.

Response: Indeed, this is a broad topic. We considered the comparison of virulence mechanisms of intracellular bacteria and parasites to viruses for manipulation of the selected organelles as novel topic of our review. The specific manipulation of a specific organelle by just bacteria has been focus of other reviews, while reviews with cross-comparison of virulence strategies are sparsely available. We selected membrane-bound organelles that are remodeled and/or repurposed by the action of the intracellular pathogen. Regarding the pathogens, focus was given to those serving as model organisms with sufficiently in-depth experimental analyses.

In the version I received, there were a significant number of references that were not formatted:

 (Error! Reference source not found.)

Response: We are sorry for this. It appears that the version of the manuscript provided to reviewers had some formatting issues. We apologize if this caused inconvenience, but this is probably an error of the editorial system. Please note that all Figures have been referred to in the text, and that citations were correctly formatted in the version we provided.

Round 2

Reviewer 1 Report

The authors have adequately addressed all of my previous concerns.

Author Response

Thank you!

Reviewer 2 Report

My opinion has not changed with the author's response.

I do not agree with the assertion: "Regarding the pathogens, focus was given to those serving as model organisms with sufficiently in-depth experimental analyses." Taking one example, poxviruses are replicated with a combination of ER and Golgi-derived membranes. This is a striking and well-researched example of pathogen engagement with these organelles. I believe that most researchers working on bacterial or viral pathogens will find their own fields absent or fleetingly described. I believe that a review that focused on a single organelle or single pathogen across organelles would be able effectively cover the topic.

I don't have a tracked changes version so it is not trivial for me to see what changes have been made that allegedly address my suggestions.

Author Response

My opinion has not changed with the author's response.

I do not agree with the assertion: "Regarding the pathogens, focus was given to those serving as model organisms with sufficiently in-depth experimental analyses." Taking one example, poxviruses are replicated with a combination of ER and Golgi-derived membranes. This is a striking and well-researched example of pathogen engagement with these organelles. I believe that most researchers working on bacterial or viral pathogens will find their own fields absent or fleetingly described. I believe that a review that focused on a single organelle or single pathogen across organelles would be able effectively cover the topic.

Response: We did not intend to provide a review focussing on manipulation of one organelle only, or only on mechanisms deployed by one pathogen only. There is already a sufficienlty high number of reviews with focus on a specific pathogen. Changing the concept of the review at this stage would require virtually starting from scratch, which we do not consider as appropriate at this stage.  Also, we do not intent to be complete in coverage of host cell organelles and intracellular pathogens. Yet we consider this review to be valuable especially for those readers which are not working on mechanisms of a specific pathogen. We agree that for specialists, more focussed reviews will be of interest, but this is not intended here.

I don't have a tracked changes version so it is not trivial for me to see what changes have been made that allegedly address my suggestions.

Response: Please note that we have provided a 'changes tracked' version of the manuscript as suppl. material with the revision. This should enable comparing the versions.

Round 3

Reviewer 2 Report

I certainly understand the authors comments that addressing my issues would mean pretty much starting from scratch.

My position hasn't changed.